Nickel mine soil is a potential source for soybean plant growth promoting and heavy metal tolerant rhizobia

Liu Han 1
Cui Yongliang 2
Zhou Jie 1
Penttinen Petri 1
Liu Jiahao 1
Zeng Lan 1
Chen Qiang 1
Gu Yunfu 1
Zou Likou 1
Zhao Ke 1
Xiang Quanju 1
Yu Xiumei xiumeiyu@sicau.edu.cn 1
1 College of Resources, Sichuan Agricultural University , Chengdu , Sichuan , China
2 Sichuan Provincial Academy of Natural Resource and Sciences , Chengdu , Sichuan , China
Moyer Craig
Electronic publication date: 2022 Apr 21
Publication date: 2022
Volume: 10
Electronic Location ID: e13215
Received 2022 Jan 20; Accepted 2022 Mar 13
Copyright: ©2022 Liu et al.
Copyright year: 2022
Copyright holder: Liu et al.
License: This is an open access article distributed under the terms of the Creative Commons Attribution License, which permits unrestricted use, distribution, reproduction and adaptation in any medium and for any purpose provided that it is properly attributed. For attribution, the original author(s), title, publication source (PeerJ) and either DOI or URL of the article must be cited.
License URL: https://creativecommons.org/licenses/by/4.0/

Keywords: Rhizobia, Nickel mine soil, Soybean, Plant growth promoting, Diversity, Heavy metal tolerance

Funding: National Natural Science Foundation of China 31872696 Key Research Project of Sichuan Province 2021YFS0293 This research was supported by the National Natural Science Foundation of China (grant number 31872696) and the Key Research Project of Sichuan Province (No. 2021YFS0293). The funders had no role in study design, data collection and analysis, decision to publish, or preparation of the manuscript.

==============================
Mine soil is not only barren but also contaminated by some heavy metals. It is unclear whether some rhizobia survived under extreme conditions in the nickel mine soil. Therefore, this study tries to isolate some effective soybean plant growth promoting and heavy metal resistant rhizobia from nickel mine soil, and to analyze their diversity. Soybean plants were used to trap rhizobia from the nickel mine soil. A total of 21 isolates were preliminarily identified as rhizobia, which were clustered into eight groups at 87% similarity level using BOXA1R-PCR fingerprinting technique. Four out of the eight representative isolates formed nodules on soybean roots with effectively symbiotic nitrogen-fixing and plant growth promoting abilities in the soybean pot experiment. Phylogenetic analysis of 16S rRNA, four housekeeping genes (atpD-recA-glnII-rpoB) and nifH genes assigned the symbiotic isolates YN5, YN8 and YN10 into Ensifer xinjiangense and YN11 into Rhizobium radiobacter, respectively. They also showed different tolerance levels to the heavy metals including cadmium, chromium, copper, nickel, and zinc. It was concluded that there were some plant growth promoting and heavy metal resistant rhizobia with the potential to facilitate phytoremediation and alleviate the effects of heavy metals on soybean cultivation in nickel mine soil, indicating a novel evidence for further exploring more functional microbes from the nickel mine soil.

Introduction

Heavy metal contamination in mining related soils affects both the mining site and the surrounding environment. Heavy metals in soil can originate from natural minerals, yet anthropogenic activities are the main source (Lebrazi & Fikri-Benbrahim, 2018). Heavy metal contamination is a risk to food security, ecological environment, and even to human health through bioaccumulation in the food chain (Zhang et al., 2012; Zhou et al., 2013; Long et al., 2021; Qin et al., 2021; Tauqeer, Turan & Iqbal, 2022). Therefore, areas with severe heavy metal-contaminations need to be remediated before being used for the cultivation of crops, and the selected crops should not accumulate contaminants when growing on the slightly-contaminated areas. Remediating the contaminated soils requires efficient and economical methods such as bioremediation, which is considered eco-friendly, secondary contamination-free, and suitable for non-point source contamination (Shao et al., 2020; Zhang et al., 2020; Yu et al., 2021).

Phytoremediation, especially in-situ enhanced phytoremediation, is widely used for the bioremediation of heavy metal-contaminated soil (Thakare et al., 2021). The growth of plants in contaminated soil can be facilitated by utilizing the biological nitrogen fixation (BNF) ability of legume-rhizobia symbionts (Hao et al., 2014; Yu et al., 2017; Wang et al., 2019; Salmi & Boulila, 2021). For example, soybean (Glycine max L. Merrill) is applicable in remediating heavy metal-contaminated sites (Li et al., 2019). In the symbiosis, rhizobia induce the formation of nodules on the roots of the host plant. Inside the nodules, rhizobia fix atmospheric nitrogen into ammonia which serves as a N source for the legume (Lindstrom & Mousavi, 2020; Wang et al., 2020). Inoculation of effective N fixing rhizobial strains leads to the growth promotion of legumes (Catroux, Hartmann & Revellin, 2001). It has been proposed that strains suitable for legume-rhizobia phytoremediation can be isolated from the contaminated sites (Limcharoensuk et al., 2015; Fan et al., 2016; Balakrishnan et al., 2017; Dhuldhaj & Pandya, 2020). Rhizobia include strains with heavy metal resistance and are capable to promote plant growth under heavy metal stress (Grandlic, Palmer & Maier, 2009; Fan et al., 2018b). It showed that a copper-resistant S. meliloti strain promoted the growth of alfalfa under copper stress (Duan et al., 2019). Rhizobial strains resistant against several heavy metals have the potential to be applied in the in-situ bioremediation of soils contaminated with multiple heavy metals (Grandlic, Palmer & Maier, 2009; Abd-Alla et al., 2012; Yu et al., 2014; Hao et al., 2015; Yu et al., 2017; Ke et al., 2021). However, indigenous rhizobia resources that could be applied in in situ phytoremediation are still scarce in Southwest China.

We hypothesized that heavy metal-contaminated soil could be a putative source for such strains. Thus, soybean plants were used to trap rhizobia from nickel mine soil in Xichang, Sichuan Province, China. The isolates were identified using molecular methods, and soybean growth-promoting abilities and heavy metal resistance of these strains were tested. The results provide better understanding of the potential of using indigenous microbial resources for the alleviation of heavy metal contamination.

Materials and Methods

Sampling and soil analysis

Soil samples were collected from a nickel mine (26°46′54.2″N, 102°06′53.2″E) in Xichang, Sichuan Province, China. Three sampling sites spaced 50 to 100 m apart were randomly selected within the area. From the sampling points, five topsoil (0–20 cm) subsamples spaced at least 5 m apart were collected and mixed to make one composite sample per site. The composite samples were quartered and stored on ice before being taken to the laboratory. The soil samples were ground, passed through a 2 mm nylon sieve, and air-dried. Soil water content in the fresh soil was determined by measuring the weight difference after soil samples had been dried at 105 °C for eight hours. Soil pH was measured using a PHS-3C pH meter (Shanghai Yoke, Shanghai, China) in a 2.5:1 water-soil slurry which had been left to settle overnight. Soil organic carbon content was determined using the K2Cr2O7-H2SO4 method (Schumacher, 2002). Total nitrogen, phosphorus and potassium contents in the soil samples were determined using Kjeldahl method (Kjeltec 8400, FOSS, Sweden), Mo-Sb colorimetric method (WFJ2100, UNICO, China) and flame spectrophotometry (FP6410, Shanghai Precision & Scientific, China), respectively (Murphy & Riley, 1962; Page, Miller & Keeney, 1982; Yu et al., 2021). Available nitrogen content was determined using the alkali N-proliferation method; soil available phosphorus and potassium were extracted with sodium bicarbonate solution and NH4AC solution, respectively, and measured using the previously described method (Wu et al., 2021). The contents of heavy metals were determined after digestion using mixed acid (HNO3: HCLO4 = 3: 1) using inductively coupled plasma optical emission spectrometry (ICP-OES; IRIS Intrepid II; Thermo Electron Corporation, Waltham, MA, USA).

Trapping and isolation of rhizobia

The indigenous rhizobial strains in the nickel mine soil were trapped using soybean cultivar Nandou No.12 bred by Nanchong Institution of Agricultural Sciences, Sichuan, China. The soybean seeds were sterilized by dipping into 95% alcohol for 3 min and 1% HgCl2 for 5 min, followed by rinsing with sterile water (Yu et al., 2017). The soybean seeds were germinated in a pot filled with sterilized moist vermiculite in the dark for 24 h, and transplanted into pots filled with soil collected from the nickel mine site (SN). The soybean plants were harvested after 90 days. Three root nodules per soybean plant were selected and sterilized using the above-mentioned methods. The nodules were incubated on beef extract peptone agar at 28 °C, and the surface sterilization was considered successful when no colonies appeared in 24 h. After that, nodules were crushed in plastic tubes and inoculated on yeast-extract mannitol agar (yeast extract 1.5 g L−1, mannitol 1.0 g L−1, K2HPO4 0.5 g L−1, MgSO4 7H2O 0.2 g L−1, NaCl 0.1 g L−1, Congo red 0.04 g L−1, agar 20 g L−1). The plates were maintained at 28 °C for 7 to 10 days. During the incubation period, single colonies were selected based on colony morphology and purified by repeated streaking (Yu et al., 2017). Purified isolates with round, plump, milky white, mucilaginous and smooth-edged colonies were examined using light microscopy after Gram staining. Gram-negative isolates with rod-shaped cells were preserved in 20% (w/v) glycerol at −80 °C.

Phylogenetic diversity analysis

The isolates were grown in YM liquid medium (yeast extract 1.5 g L−1, mannitol 1.0 g L−1, K2HPO4 0.5 g L−1, MgSO4 7H2O 0.2 g L−1, NaCl 0.1 g L−1, Congo red 0.04 g L−1) for 24 h at 28 °C in a shaking incubator, and DNA was extracted using TIANamp Bacteria DNA Kit (TIANGEN, China). The genetic diversity of the isolates was assessed using BOX-A1R PCR fingerprinting with the primer 5′-CCTCGGCAAGGACGCTGACG-3′ (Chen et al., 2014). Amplification was done in a 10 µL volume system containing 5 µL of 2 × PCR mix, 0.2 µL of the primer (10 µmol L−1), 1 µL of template DNA (50 ng mL−1), and 3.8 µL double distilled water (ddH2O). The BOX-A1R PCR thermal profile included initial denaturation at 94 °C for 3 min, 30 cycles of denaturation at 94 °C for 1 min, annealing at 52 °C for 1 min and extension at 65 °C for 8 min, and a final extension at 65 °C for 16 min. The amplified fragments (8 µL) from the amplification mixture and the 200 bp DNA ladder were separated in 2% (w/v) agarose gel with ethidium bromide at 80 V for 2.5 h, and were visualized under UV light with the patterns recorded. Based on the patterns, a BOX-A1R PCR cluster tree diagram was created using the unweighted pair group method with arithmetic averages (UPGMA) in NTSYSpc 2.1 (Yu et al., 2014).

Plant growth promotion ability test

Based on the BOX-A1R PCR fingerprints, eight representative rhizobial isolates were selected for the test of plant growth promotion ability using Leonard jars. Leonard jar and nitrogen-free nutrient solution were prepared as previously described (Kang et al., 2018; Yu et al., 2017). Soybean seeds were surface-sterilized and germinated as described above. Three sterilized seeds were transferred into the topsoil of a Leonard jar, which was then covered with a layer of approximately 3 cm moist vermiculite. When the soybean seedlings were 2–3 cm tall, the rhizobia culture in exponential phase (OD600 nm 0.6∼0.8) was inoculated around the roots, and then the topsoil was covered with a layer of 1 cm autoclaved quartz sand. The non-inoculated soybean supplemented with nitrogen-free nutrient solution was set as the negative control, while the N+ treatment with 1 g L−1KNO3 as the nitrogen source in the nutrient solution was the positive control. Each treatment and control included three replicates. The soybean plants were kept in a greenhouse with artificial light-dark cycles: 17 h light, temperature 25 °C and humidity 80% (to simulate daytime); 7 h dark, temperature 17 °C and humidity 85% (to simulate nighttime). The jars were replenished with nutrient solution when needed. The soybean plants were harvested after 55 days. The number of nodules, root length, and plant height and weight were measured. The plant samples were dried at 105 °C for 30 min to deactivate enzymes, and then at 55 °C until constant weight. The nitrogen, phosphorus, and potassium contents of the dried roots and shoots were measured.

Sequence analysis

The isolates that nodulated soybean plants were further characterized using sequence analyses. The almost full length 16S rRNA gene was amplified using the primer pair 27F/1492R (Yu et al., 2017), the housekeeping genes atpD, recA, glnII, and rpoB using primer pairs atpD 255F/ atpD 782R, recA 63F/recA 555R, glnII 12F/glnII tsR, and rpoB 454F/ rpoB 1364R, respectively (Tang et al., 2012; Zhao et al., 2014), and the nitrogen fixation gene nifH using the primer pair nifHF/nifHI (Laguerre et al., 2001). All of the primers we used in this study were listed in Table S2. Amplification was done in a 30 µL volume system containing 15 µL of 2 × PCR mix, 0.15 µL of each primer, one µL of template DNA (50 ng mL−1), and 13.7 µL ddH2O. For the amplification of 16S rRNA, atpD, recA, glnII and rpoB genes, the thermal cycling conditions included an initial denaturation at 94 °C for 3 min, 30 cycles of denaturation at 94 °C for 1 min, annealing at 55 °C for 1 min, extension at 72 °C for 2 min, and a final extension at 72 °C for 10 min. For the amplification of nifH, the thermal cycling conditions included an initial denaturation at 95 °C for 3 min, 30 cycles of denaturation at 94 °C for 1 min, annealing at 59 °C for 1 min, extension at 72 °C for 5 min, and a final extension at 72 °C for 6 min. The amplified products were sequenced by Sangon Biotech (Shanghai, China). The fragments of atpD (274 bp), recA (245 bp), glnII (413 bp) and rpoB (534 bp) were concatenated for multilocus sequence analysis (MLSA). The sequences were matched against reference sequences of rhizobia in the GenBank of NCBI database using BLAST. The reference sequences of different rhizobia species we used for the phylogenetic analysis of the three genes (16SrRNA, housekeeping genes, and nifH) were separately listed in Tables S3, S4, and S5. The sequences of the isolates and the reference sequences were subjected to phylogenetic analysis using neighbor joining method in MEGA7.0. The phylogenetic trees were bootstrapped with 1,000 replications for each sequence to evaluate the reliability of the tree topologies (Saitou & Nei, 1987).

Heavy metal resistance test

For assessing the heavy metal-resistance ability of the isolates, 10 g L−1 stock solutions of Cd2+ (CdCl2 2.5H2O), Cr6+ (K2Cr2O7), Cu2+ (CuCl2 2H2O), Ni2+ (NiSO4), and Zn2+ (ZnSO4 7H2O) were prepared by dissolving the salts in ultrapure water. The isolates were grown in 5 mL YM liquid medium with 0, 4, 8, 12, 16, 20, 40, 60, 80, 100 mg L−1 of each metal in an orbital shaker at 28 °C for 72 h, after which the optical density (OD600 nm) of the cultures were measured using a spectrophotometer (UV-3300; Shanghai MAPADA, Shanghai, China) (Abd-Alla et al., 2012).

The minimum inhibitory concentrations (MIC) were determined by comparing the OD600 nm value of the cultures spiked with metals to those without metals. MIC was defined as the lowest metal concentration resulting in a visually observable decrease in the growth curve of the isolates. Minimum lethal concentration (MLC) was defined as the lowest metal concentration resulting in an OD600 nm value lower than 0.1.

Statistical analysis

Statistical differences in soil properties and soybean growth parameters were tested using one-way ANOVA and Fisher’s protected LSD (least significant difference) test at P ≤ 0.05 in IBM SPSS statistics 22.0. The differences in MIC and MLC values were not tested due to their zero variance.

Results

Isolation of rhizobia from nickel mine soil

The soil in the nickel mining area was slightly acidic (pH 6.35), and the organic matter content was low (1.07%). The contents of total N, P and K were 261.35, 479.85, and 5869.73 mg kg−1, respectively, and 15.08%, 2.93%, and 0.38% of them were available fragments (Table 1). Both nickel and lead contents were approximately 20 mg kg−1, and cadmium, chromium, copper, iron, manganese and zinc were also detected (Table 1).

Table 1 The physicochemical properties and heavy metal contents of the nickel mine soil collected in Xichang, Sichuan Province, China.

Properties	pH	WC (%)	OM (%)	TN (mg kg −1 )	AN (mg kg −1 )	TP (mg kg −1 )	AP (mg kg −1 )	TK (mg kg −1 )	AK (mg kg −1 )	
Average value	6.35 ± 0.01	7.71 ± 0.00	1.07 ± 0.06	261.35 ± 22.74	39.21 ± 3.04	479.85 ± 0.00	140.69 ± 1.44	5869.73 ± 0.05	22.68 ± 2.91	
Properties	Ni (mg kg −1 )	Cu (mg kg −1 )	Cr (mg kg −1 )	Pb (mg kg −1 )	Mn (mg kg −1 )	Fe (mg kg −1 )	Zn (mg kg −1 )	Cd (mg kg −1 )		
Average value	19.69 ± 0.44	9.79 ± 0.10	12.78 ± 0.08	20.13 ± 0.03	8.57 ± 0.04	744.20 ± 4.10	80.07 ± 1.04	0.46 ± 0.01		
Notes.

WC water content

OM soil organic matter content

TN soil total nitrogen content

AN soil available nitrogen content

TP soil total phosphorus content

AP soil available phosphorus content

TK soil total potassium content

AK soil available potassium content

Ni nickel

Cu copper

Cr chromium

Pb lead

Mn manganese

Fe iron

Zn zinc

Cd cadmium

The values are expressed as mean standard deviation (n = 3).

As a result of trapping rhizobial strains from the nickel mine soil using soybean as the host plant, 21 isolates were preliminarily identified as rhizobia based on colony morphology, microscopic examination of cell shape, and Gram staining. The BOXA1R-PCR fingerprinting (Fig. 1) of the isolates showed that sixteen distinct fingerprint patterns were found, indicating that these isolates from nickel mine soil were genetically diverse. The isolates were clustered into 8 groups at 87% similarity level.

Figure 1 Dendrogram of the BOX-A1R PCR fingerprints of 21 isolates trapped by soybean from the nickel mine soil collected in Xichang, Sichuan Province, China.

Plant growth promotion ability of selected isolates

Based on the fingerprinting, we selected 8 isolates (YN1, YN5, YN8, YN10, YN11, YN13, YN27, YN30) for the plant growth promotion ability test. Only four isolates (YN5, YN8, YN10, YN11) nodulated soybean plants, with nodule numbers ranging from 50 to 54 per plant (mean value of six plants). The biomass of soybeans inoculated with the four symbiotic isolates (YN5, YN8, YN10, YN11) and isolate YN13 was significantly higher than that in the N− treatment (P = 0.00) (Fig. 2A). The biomass of soybeans inoculated with isolate YN8 was on the same level as in the N+ treatment. The shoots of soybeans inoculated with the 4 symbiotic isolates (YN5, YN8, YN10, YN11) and isolate YN1 were significantly longer than those in the N− treatment (P = 0.00, 0.01, 0.00, 0.00, 0.00). The roots of soybeans inoculated with the symbiotic isolates YN8, YN10, and YN11 were significantly longer than those in the N− treatment (P = 0.00) (Fig. 2B). The shoot N content of soybean plants inoculated with the symbiotic strains was at the same level as in the N+ treatment, and the shoot N content of soybeans inoculated with the symbiotic strains of YN5, YN8, YN10, and YN11 as well as the isolate YN1 was significantly higher than that in the N− treatment (P = 0.00) (Fig. 3A). The root N content of inoculated soybean plants was higher than that in the N− treatment (Fig. 3A). The root P content of soybeans inoculated with the symbiotic isolates and isolate YN30 was lower than that in the N− treatment (Fig. 3B). The shoot K content of soybeans inoculated with the isolate YN10 was higher than that in the N− treatment (Fig. 3C). The root K content of soybeans inoculated with the isolates YN5, YN8, YN10, YN13, YN27 and YN30 was higher than that in the N− treatment (Fig. 3C).

Figure 2 Biomass (A) and shoot and root lengths (B) of soybean plants inoculated with isolates trapped from the nickel mine soil collected in Xichang, Sichuan Province, China.

Different letters above the bars indicate significant statistical difference at p < 0.05 (n = 3). Comparison between shoot lengths is in normal font and comparison between root lengths is in bold.

Figure 3 Nitrogen (A), phosphorus (B) and potassium (C) contents in the shoots and roots of soybean plants inoculated with the isolates trapped from the nickel mine soil collected in Xichang, Sichuan Province, China.

Different letters above bars indicate significant statistical difference at p < 0.05 (n = 3). Comparison between shoot lengths is in normal font and comparison between root lengths is in bold.

Sequence analyses of symbiotic isolates

As a result of the sequence analysis of the almost full length 16S rRNA gene, the symbiotic isolates were assigned to the genera Ensifer (formerly designated as Sinorhizobium) and Rhizobium (Fig. 4). The isolates YN5, YN8, and YN10 were respectively grouped together with the type strains E. fredii USDA205, E. americanum CFNEI156 and E. xinjiangense CCBAU110, and isolate YN11 with R. radiobacter ICMP 5856 (formerly Agrobacterium tumefaciens ICMP 5856). Based on the multilocus sequence analysis (MLSA) of the concatenated fragments of atpD (274 bp), recA (245 bp), glnII (413 bp) and rpoB (534 bp), the isolates YN5, YN8, and YN10 were identified as E. xinjiangense strains, and YN11 as Rhizobium radiobacter strain (Fig. 5). The nifH sequences from YN5, YN8, and YN10 were 100% similar with those from E. fredii CCBAU23314 and E. xinjiangense CCBAU110, and the nifH sequence from YN11 with that from R. radiobacter (Fig. 6).

Figure 4 16S rRNA gene phylogeny of the soybean symbiotic isolates from the nickel mine soil collected in Xichang, Sichuan Province, China (in bold), and reference strains.

GenBank accession numbers are in parentheses. Bootstrap values >50% are shown at the branch points. The scale bar denotes 5% substitutions per site.

Figure 5 Multilocus sequence analysis (MLSA) of the soybean symbiotic isolates from the nickel mine soil collected in Xichang, Sichuan Province, China (in bold), and reference strains.

MLSA was done using the concatenated fragments of atpD (274 bp), recA (245 bp), glnII (413 bp) and rpoB (534 bp). GenBank accession numbers are in parentheses. Bootstrap values >50% are shown at the branch points. The scale bar denotes 5% substitutions per site.

Figure 6 nifH phylogeny of the soybean symbiotic isolates from the nickel mine soil collected at Xichang, Sichuan Province, China (in bold), and reference strains.

GenBank accession numbers are in parentheses. Bootstrap values >50% are shown at the branch points. The scale bar denotes 5% substitutions per site.

Heavy metal resistance ability of symbiotic isolates

In general, E. xinjiangense YN5 and R. radiobacter YN11 tolerated higher levels of heavy metals compared to E. xinjiangense YN8 and YN10 (Fig. 7, Table S1). For Cd2+, the MIC and MLC values of E. xinjiangense YN5 and R. radiobacter YN11 were the highest. For Cr6+, the MLC of all the strains was 16 mg L−1. For Cu2+, the MIC and MLC values of E. xinjiangense YN8 were the lowest. For Ni2+, the MLC values of YN5 and YN11 were higher than those of YN8 and YN10. For Zn2+, the MLC value of E. xinjiangense YN5 was 300 mg L−1, i.e., over three times higher than that of R. radiobacter YN11 and over 18 to 75 times higher than those of YN11 and YN8, respectively.

Figure 7 The growth of the soybean symbiotic isolates from the nickel mine soil collected in Xichang, Sichuan Province, China, at increasing concentrations of Cd2+ (A), Cr6+ (B), Cu2+ (C), Ni2+ (D), and (E) Zn2+.

Minimum inhibitory concentrations (MIC) are indicated with green and minimum lethal concentrations (MLC) with red.

Discussion

We trapped rhizobia from nickel mine soil using soybean plants in Xichang, Sichuan Province, China, to find effectively plant growth promoting and heavy metal resistant strains for the enhancement of phytoremediation of heavy metal contaminated soil and for the promotion of soybean growth on slightly contaminated farmland. The low organic matter, N, P and K implied that the soil was barren (Zhang et al., 2012; Wu et al., 2021), yet the rhizobia-trapping plants were nodulated, and the isolates from nodules were diverse based on the BOXA1R-PCR fingerprints. However, when inoculated on soybeans, only four out of the eight representative isolates formed nodules on the roots. Similar to rhizobia isolated from Glycyrrhiza spp. (Li et al., 2012), the four non-nodulating isolates may have been sporadic symbionts or other endophytes that had entered the nodules along with a genuine symbiont. Similar to the model inoculant of soybean, Bradyrhizobium diazoefficiens USDA110 (Sibponkrung et al., 2020), inoculation with the symbiotic isolates resulted in over two times higher biomass than in the non-inoculated control; the higher biomass was accompanied with higher shoot nitrogen content. In addition, even the non-nodulating isolates showed some plant growth promoting abilities. Especially, inoculation with all the representative isolates resulted in higher root N content than in the nitrogen-free control. In most of the inoculated plants, the increase in root N content was accompanied with lower P content.

As a host plant, soybean is promiscuous and may be nodulated with both fast- and slow-growing rhizobia (Chen et al., 2021). Based on the 16S rRNA gene analysis, three of our isolates were identified as Ensifer americanum, E. fredii or E. xinjiangense, i.e., species with closely related 16S rRNA genes (Peng et al., 2002; Wang et al., 2013). Further analysis using MLSA of the four housekeeping genes assigned the symbiotic isolates into the fast-growing rhizobial species Ensifer xinjiangense and Rhizobium radiobacter, strains of which have been identified as plant growth promoting symbionts of soybean plants (Peng et al., 2002; Iturralde et al., 2019). To our knowledge, neither E. xinjiangense (formerly Sinorhizobium xinjiangense) nor R. radiobacter (formerly Agrobacterium tumefaciens) strains have been applied as a single-inoculant plant growth promoter in bioremediation, yet co-inoculation of an A. tumefaciens strain with S. meliloti promoted the growth of Medicago lupulina under Cu and Zn stresses (Jian et al., 2019). R. radiobacter is traditionally considered as a plant pathogen and is a free-living nitrogen fixer (Kanvinde, Sastry & Microbiology, 1990). In our study, the amplification and sequencing of the nifH gene, which encodes nitrogenase iron protein, showed that both the Ensifer strains and R. radiobacter YN11 had the genetic potential for nitrogen fixation. The nodulation and plant growth promotion capacity of R. radiobacter YN11 added to the growing body of evidence that when carrying the nodulation genes, Rhizobium (Agrobacterium) clade strains can be legume-nodulating symbionts (Cummings et al., 2009).

In the soil from mining areas, the concentrations of heavy metals vary considerably from below the background values to hundreds of times higher than the average values for all soils (Li et al., 2014). Bacteria inhabiting the heavy metal-contaminated sites include legume nodulating strains with high tolerance against the metals (Mohamad et al., 2017). In our study, the symbiotic strains showed varied heavy metal resistance; E. xinjiangense YN5 outperformed the other E. xinjiangense isolates and the resistance levels of R. radiobacter YN11 fell in-between. Compared to the rhizobia isolated directly from V-Ti magnetite mine tailing soil and those from the nodules Robinia pseudoacacia in a Pb-Zn mining area (Yu et al., 2014; Fan et al., 2018a; Fan et al., 2018b), our isolates tolerated lower levels of Cd2+ and Cu2+. One possible explanation is the level of contamination on the sites where the strains were isolated; the V-Ti magnetite and Pb-Zn mining areas were seriously contaminated (Yu et al., 2014; Fan et al., 2018a), but only Zn content in the nickel mine soil was higher than in the background value for soils in China (Li et al., 2014). The levels of heavy metals tolerated are approximately 10 to 100 times lower in liquid medium than on solid medium (Hassen et al., 1998). It is also important to take into account the different testing methods for the Zn tolerance of E. xinjiangense YN5. The V-Ti magnetite and Pb-Zn mining area isolates were tested on solid media (Yu et al., 2014; Fan et al., 2018a) but our isolates in liquid medium, yet E. xinjiangense YN5 tolerated a higher level of Zn2+.

Conclusions

Our study used soybean pot experiment to trap 21 rhizobia strains from nickel mine soil. As a result, we selected three Ensifer xinjiangense strains (YN5, YN8, and YN10) and one Rhizobium radiobacter (YN11) with good nitrogen fixing ability, which can significantly improve the soybean plant height, root length, and biomass yield. Moreover, these four strains carried the symbiotic gene nifH that can encode dinitrogenase reductase enzyme, which further confirmed their abilities to form root nodules and fix nitrogen. E. xinjiangense YN5 and R. radiobacter YN11 tolerated higher levels of heavy metals than E. xinjiangense YN8 and YN10. Taken together, the results showed that the nickel mine soil is a potential source for plant growth promoting rhizobia strains, which could be applied as indigenous inoculants in the phytoremediation of slightly contaminated farmland and in the alleviation of the adverse effects of heavy metals on soybean cultivation.

Supplemental Information

Table S1 Heavy metal resistance ability of the soybean symbiotic Ensifer xinjiangense (YN5, YN8, YN10) and Rhizobium radiobacter (YN11) isolates from the nickel mine soil collected from Xichang, Sichuan Province, China. MIC, minimum inhibitory concentr

Note: The differences in MIC and MLC values were not tested due to their zero variance.

Click here for additional data file.

Table S2 Primer sequences used in this study

Click here for additional data file.

Table S3 Reference strains used in 16S rRNA phylogenetic tree

Click here for additional data file.

Table S4 Reference strains used in housekeeping genes-MLSA phylogenetic tree

Click here for additional data file.

Table S5 Reference strains used in nifH phylogenetic tree

Click here for additional data file.

Supplemental Information 6 Raw data

Nickel mine soil basic properties, soybean growth, and the resistant ability of symbiotic rhizobia strains against five heavy metals.

Click here for additional data file.

Supplemental Information 7 The BOX-A1R recording information of all rhizobia isolates that isolated from nickel mine soil

Click here for additional data file.

Supplemental Information 8 Partial sequences of four symbiotic rhizobia strains

Click here for additional data file.

Supplemental Information 9 GenBank accession numbers of four symbiotic rhizobia strains

Click here for additional data file.

We thank Dr Xia Kang (School of Life Sciences, University of Dundee, United Kingdom) for English language editing.

Additional Information and Declarations

Competing Interests

Author Contributions

Data Availability

The authors declare there are no competing interests.

Han Liu conceived and designed the experiments, performed the experiments, analyzed the data, prepared figures and/or tables, authored or reviewed drafts of the paper, and approved the final draft.

Yongliang Cui and Xiumei Yu conceived and designed the experiments, analyzed the data, prepared figures and/or tables, authored or reviewed drafts of the paper, and approved the final draft.

Jie Zhou performed the experiments, prepared figures and/or tables, and approved the final draft.

Petri Penttinen analyzed the data, prepared figures and/or tables, authored or reviewed drafts of the paper, and approved the final draft.

Jiahao Liu and Lan Zeng performed the experiments, prepared figures and/or tables, and approved the final draft.

Qiang Chen, Yunfu Gu, Likou Zou, Ke Zhao and Quanju Xiang analyzed the data, authored or reviewed drafts of the paper, and approved the final draft.

The following information was supplied regarding data availability:

The sequences are available at NCBI GenBank: OM288131–OM288134, OM669835–OM669838, OM669843–OM669846, OM669851–OM669834, OM669839–OM669842, OM669847–OM669850.

The raw data, nickel mine soil basic properties, soybean growth, and heavy metal resiatance ability of rhizobia isolates are available in the Supplementary File.

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
