# Peer review of "Nickel mine soil is a potential source for soybean plant growth promoting and heavy metal tolerant rhizobia"

_PeerJ, doi:10.7717/peerj.13215_

## Round 0.1 · original submission · Major Revisions

I agree with the reviewers that some revisions are in order before this manuscript can be considered further for publication. Please carefully address each of the reviewers’ comments when resubmitting your manuscript. The primary concern was that there were several grammatical issues that need attention.

Reviewer 1 has suggested that you cite specific references. You are welcome to add it/them if you believe they are relevant. However, you are not required to include these citations, and if you do not include them, this will not influence my decision.

Reviewer 1 ·

Basic reporting

Comments for Author,

In this paper, The Author tries to explore the " Nickel mine soil is a potential source for soybean plant growth-promoting and heavy metal tolerant rhizobia". The paper is the scope of Peerj. However,
I have some comments before accepting this paper.

Comments for Author,

These sentences need more related papers. to improve please support it. " Heavy metal
contamination is a risk to food security, ecological environment, and even to human health
through bioaccumulation in the food chain" a- https://doi.org/10.1007/978-3-030-89984-4_19

This below sentences make no sense;

In this study, through the cultivation experiment of soybean plant in nickel mine soil, we
aimed to find effective plant growth promoting and heavy metal resistant strains to facilitate
phytoremediation of heavy metal contaminated soil.

Please rewrite the above sentences.

Please give the coordinate of soil that the Authors collected.

Plant growth ability test section should be reduced. There are some confusion

Best Regards

Experimental design

The design of the experiment was acceptable.

Validity of the findings

The results are reasosable.

·

Basic reporting

In the current study, authors have isolated plant growth promoting symbiotic strains from nickel mine soils with the idea of using such isolates for phytoremediation. The manuscript is well written with detailed description of the methods and proper interpretation of the results.

Few minor suggestions/corrections are detailed in the edited version of the manuscript. Please address them.

Experimental design

All the research questions are well defined and authors have conducted all the necessary experimentals to support their study.

Validity of the findings

The result are well represented and all the necessary raw files were provided.
However, it would be good to include the sequencing files as a word or text file in a zip format so that it is easily accessible.
Authors should include title and figure legends or footnotes for all the tables and figures wherever applicable along with the abbreviations.

---

## Round 0.2 · accepted · Accept

I concur with the reviewers that significant enhancements have been made to the manuscript allowing it to now be accepted for publication. This is especially the case with the revised figure legends.

Reviewer 1 ·

Basic reporting

The Author improved the paper carefully, Now the paper could be accepted for publication.

Best Regards

Experimental design

Well

Validity of the findings

Well

·

Basic reporting

Authors have addressed all the comments and suggestions made in earlier review report. All the corrections are satisfactory and the manuscript can be accepted in its current form.

Experimental design

No comments

Validity of the findings

No comments